# Large-Scale Defect Clusters with Hexagonal Honeycomb-like Arrangement in Ammonothermal GaN Crystals

**DOI:** 10.3390/ma15196996

**Published:** 2022-10-09

**Authors:** Lutz Kirste, Thu Nhi Tran Thi Caliste, Jan L. Weyher, Julita Smalc-Koziorowska, Magdalena A. Zajac, Robert Kucharski, Tomasz Sochacki, Karolina Grabianska, Malgorzata Iwinska, Carsten Detlefs, Andreas N. Danilewsky, Michal Bockowski, José Baruchel

**Affiliations:** 1Fraunhofer Institute for Applied Solid State Physics, Tullastraße 72, D-79108 Freiburg, Germany; 2European Synchrotron Radiation Facility, 71 Avenue des Martyrs, F-38043 Grenoble, France; 3Institute of High-Pressure Physics, Polish Academy of Sciences, Sokołowska 29/37, 01-142 Warsaw, Poland; 4Faculty of New Technologies and Chemistry, Military University of Technology, ul. Gen. Sylwestra Kaliskiego 2, 00-908 Warsaw, Poland; 5Crystallography, Institute of Geo- and Environmental Sciences, Albert-Ludwigs-Universität Freiburg, Hermann-Herder-Straße 5, D-79104 Freiburg, Germany; 6Center for Integrated Research of Future Electronics, Institute of Materials and Systems for Sustainability, Nagoya University, C3-1 Furo-cho, Chikusa-ku, Nagoya 464-8603, Japan

**Keywords:** gallium nitride (GaN), X-ray Bragg diffraction imaging, rocking curve imaging, defect selective etching, honeycomb defect, limited size subgrain boundary, geometrically necessary dislocation, Borrmann effect

## Abstract

In this paper, we investigate, using X-ray Bragg diffraction imaging and defect selective etching, a new type of extended defect that occurs in ammonothermally grown gallium nitride (GaN) single crystals. This hexagonal “honeycomb” shaped defect is composed of bundles of parallel threading edge dislocations located in the corners of the hexagon. The observed size of the honeycomb ranges from 0.05 mm to 2 mm and is clearly correlated with the number of dislocations located in each of the hexagon’s corners: typically ~5 to 200, respectively. These dislocations are either grouped in areas that exhibit “diameters” of 100–250 µm, or they show up as straight long chain alignments of the same size that behave like limited subgrain boundaries. The lattice distortions associated with these hexagonally arranged dislocation bundles are extensively measured on one of these honeycombs using rocking curve imaging, and the ensemble of the results is discussed with the aim of providing clues about the origin of these “honeycombs”.

## 1. Introduction

Gallium nitride (GaN) wafers with low defect density are the substrate of choice for demanding GaN-based optoelectronic and electronic device structures. These include lateral high electron mobility transistors (HEMTs) and vertical field-effect transistors (FETs) as well as laser diodes (LDs) [1,2,3]. A suitable method to produce high-quality GaN crystals with low defect density is the ammonothermal approach [4,5,6], a solvothermal growth technique from a solution where a sufficient solubility of GaN is achieved by supercritical ammonia and mineralizers under high pressure. The ammonothermal process for growth of GaN (Am-GaN) is the ammonia counterpart of the hydrothermal crystallization of quartz from supercritical water. Such an approach is characterized by a high reproducibility. Several dozen crystals can be produced simultaneously in an autoclave in one growth run. So far, the ammonothermal growth method is the technique best suited to mass production that yields GaN crystals with the lowest threading dislocation density (TDD < 1 × 10^4^ cm^−2^) [7].

Recently, Kirste et al. demonstrated the high crystalline quality of entire 1.8-inch Am-GaN substrates by Lang-technique X-ray Bragg diffraction imaging (historically called “X-ray topography”) [8]. The main contrast mechanism for the observation of the defects was, in this case, the anomalous transmission (“Borrmann effect”) predicted by the dynamical theory of diffraction (see, for instance, Authier [9]). The occurrence of the Borrmann effect in the studied substrates constitutes clear evidence of the high crystalline perfection of Am-GaN grown by a native seed approach. Using this laboratory X-ray Bragg diffraction imaging technique, not only microscopic defects such as threading dislocations (TDs), but also macroscopic defects, e.g., dislocation clusters due to preparation insufficiency, traces of facet formation, growth bands, dislocation walls, and dislocation bundles, were detected [8].

This previous study pointed out the presence, within the Am-GaN crystals, of a new type of defect cluster characterized by its hexagonal shape, macroscopic (up to 2 mm) size, and locally high defect density. We show in the present publication that this “honeycomb-like” defect cluster is mainly composed of six dislocation bundles (DBs) located in the corners of the hexagon. In order to characterise these defect clusters, several Bragg diffraction imaging techniques were combined with defect selective etching. The detailed examination of this newly discovered “honeycomb defect” (HCD) in Am-GaN crystals is the subject of the present work.

## 2. Samples and Experimental Techniques

### 2.1. Investigated Ammonothermal GaN Crystals

The investigated GaN crystals were grown by basic ammonothermal crystal growth technology in Ni-based alloy high-pressure autoclaves in a temperature range of 500–600 °C and ammonia pressure of 0.3–0.4 GPa. The growth principle is based on dissolving a polycrystalline GaN feedstock in one zone of the high-pressure autoclave filled with supercritical ammonia and transporting the dissolved material to the second zone, where the solution is supersaturated and the crystal growth of GaN takes place on native seeds [5,6,10]. Owing to a negative temperature coefficient of solubility, the so-called retrograde solubility, the chemical transport of GaN is directed from the low-temperature solubility zone to the high-temperature crystallization zone [11]. The growth process is complex: the first step aims to produce, from smaller seeds, GaN crystals with sufficient sizes. These are then used as new seeds for preparing substrates, which are obtained by applying proper wafering procedures. Further details of the basic ammonothermal growth and wafering process can be found elsewhere [5,6,10,12].

More than 40 Am-GaN substrates from different growth runs were investigated in this study. They included n-type material with a high oxygen donor concentration and semi-insulating material with manganese compensation. The diameter of the investigated Am-GaN (0001) *c*-plane substrates was 1 inch, 1.5 inches, 1.8 inches, or 2 inches with a thickness of 300 µm–450 µm. Both surfaces of the substrates, parallel to the (0001) and $\left(000\bar{1}\right)$ crystallographic planes, were prepared to exhibit a roughness in the 0.1 nm range (“epi-ready state”).

### 2.2. X-Ray Bragg Diffraction Imaging

The defect analysis of these low-defect density Am-GaN crystals requires characterization techniques with a large field of view and high sensitivity to crystalline distortions, able to visualize the dislocations and grain boundaries as well as to provide a measurement of the lattice distortions. X-ray Bragg diffraction imaging (historically called X-ray topography), which allows imaging defects/dislocations in bulk crystals with low dislocation density (∼10^5^ cm^−2^ and below), fulfils these requirements. This X-ray imaging method is sensitive to weak misorientations (typically a few microradian) and strains (in the 10^−6^ range), over large sample areas (~cm^2^-range), with a spatial resolution in the µm range [9,13].

The classical X-ray Bragg diffraction imaging technique is based on the analysis of the X-ray diffracted beam intensity recorded by a two-dimensional detector. The “local” intensity recorded on a given area of the detector is a function of the gradient of distortion (often associated with defects) within the volume of the crystal that contributed to the diffraction on this detector area [9]. This analysis provides very useful information about the defect structure, but this information usually remains qualitative, or semi-quantitative. A synchrotron-radiation based development of monochromatic X-ray Bragg diffraction imaging, known as rocking curve imaging (RCI) [14], provides actual quantitative information about the local lattice distortion by recording the diffracted intensity as a function of the rocking curve angle.

For the investigation of the “honeycomb-like” defects (HCDs) in Am-GaN, we combined three different X-ray Bragg diffraction imaging techniques, namely, laboratory X-ray Lang topography, synchrotron white beam X-ray topography, and synchrotron monochromatic X-ray rocking curve imaging.

*Lang X-ray topography**(L-XRT):* In 1958, Lang designed an X-ray imaging technique such that the crystal, set to be in Bragg position, and the detector are simultaneously translated in front of the incident X-ray beam defined by a ~100 µm wide vertical slit [15]. This allows producing a Bragg diffracted image of the whole crystal. Therefore, Lang topography provides an overview of the defects present in the Am-GaN wafers, which indicates the location of the HCD clusters. Images of the entire wafers were performed in projection transmission geometry using a RIGAKU XRTmicron laboratory X-ray topography camera (RIGAKU, Tokyo, Japan). The X-ray beam originates from a high-brilliance microfocus X-ray source combined with a multilayer X-ray optics, which provides Cu-Kα_1_ radiation (λ = 154.06 pm, 8.05 keV). With the used Cu-Kα_1_ radiation and a typical GaN wafer thickness of about 300 µm to 450 µm, the absorption is rather high, with µt in the 9.1–13.6 range, where µ is the linear X-ray absorption coefficient and t is the crystal thickness. This means that the conditions for the “anomalous transmission” Borrmann effect X-ray topography are met if the Am-GaN wafers exhibit a sufficient perfection. Further details on the Borrmann effect in Am-GaN can be found in the work of Kirste et al. [8] and a detailed description of the fundamentals of the Borrmann effect is given, for instance, by Borrmann [16] and Authier [9]. For the imaging of each wafer, one of the six *a*-plane $\left\{11\bar{2}0\right\}$ reflections ($11\bar{2}0$, $\bar{1}\bar{1}20$, $1\bar{2}10$, $\bar{1}2\bar{1}0$, $2\bar{1}\bar{1}0$, or $\bar{2}110$) was used. In order to include a correction for the curvature of the GaN wafers, automated Bragg angle control was applied to keep the crystals in scattering condition during the imaging scans [17]. The X-ray topographs were recorded by a high-resolution CCD camera (5.4 × 5.4 μm^2^ pixel size) with a scanning speed of 1 mm per minute. In addition to the overview images, higher resolution images of selected HCDs were recorded with an ultra-high-resolution CCD camera (2.4 × 2.4 μm^2^ pixel size) to identify details of the defect clusters.

*Synchrotron**white-beam X-ray topography (SWB-XRT):* Selected Am-GaN wafer areas with HCD clusters were investigated at the topography station at the imaging cluster of the Karlsruhe Research Accelerator (KARA) synchrotron (former name ANKA) at the Karlsruhe Institute of Technology (KIT), Germany, utilizing the polychromatic synchrotron radiation beam (“white beam”) emitted by a bending magnet. Further details of the beam line characteristics at the KARA synchrotron light source can be found elsewhere [18]. White beam topography is analogous to the Laue method: the crystal is located in the nearly parallel polychromatic beam and each family of lattice planes (hkil) selects a wavelength λ from the continuous spectrum that fulfills the diffraction condition according to Bragg’s law 2d_hkil_ sinθ_B_ = nλ, where d_hkil_ is the inter-planar spacing of the lattice planes (hkil), θ_B_ is the Bragg angle, n is the reflection order, and λ is the wavelength of the diffracted beam. In this way, a series of images (“topographs”) of the illuminated part of the sample are simultaneously recorded during a single exposure [19,20]. A dislocation induces an associated distortion field, and becomes visible through the contrast variation in the topographic image performed using planes affected by this distortion. The SWB-XRT technique is, in particular, sensitive to misorientations within the crystal: the presence of subgrains is visualized through an orientation contrast resulting from diffracted intensities originating from the various subgrains that propagate in slightly different directions.

SWB-XRT was performed using an extended parallel beam (“projection topographs”) for both the transmission and reflection geometries. LauePT software was used to set the orientation of the Am-GaN substrates on the goniometer and for the indexation of the Laue pattern [21]. For the exposures in transmission geometry, the samples were tilted by either plus or minus 12 degrees about the horizontal axis; this allowed access to the Laue spots (topographs) of each of the six types of $\left\{11\bar{2}0\right\}$ and $\left\{\bar{1}100\right\}$ prismatic lattice planes. For the above-mentioned wafer thicknesses, the µt value amounts to 7.1 and 20.5 for the type $11\bar{2}0$ reflection ($\lambda_{11\bar{2}0}≅$ 66.30 pm, 18.70 keV) and for the $\bar{1}100\;$($\lambda_{\bar{1}100\;}≅$ 99.45 pm, 12.47 keV) reflection, respectively. A sample-to-film distance of about 100 mm was used and the X-ray beam size was limited to 3 × 3 mm^2^ to avoid overlapping of the various topographs on the film-detector. The 0006 reflections were measured by means of the large-area back-reflection geometry. Here, the wafers were tilted by 7 degrees about the horizontal axis, the sample-to-film distance was again 100 mm, and the X-ray beam size was limited to 5 × 5 mm^2^. The transmission topographs were recorded on SLAVICH VRP-M high-resolution films (Geola, Vilnius, Lithuania) and were digitized afterwards. A high-resolution Excelitas PCO *4000* CCD camera with a pixel size of 2.5 × 2.5 μm^2^ (Excelitas PCO GmbH, Kelheim, Germany) was used for back-reflection topography.

*Synchrotron monochromatic rocking curve imaging (RCI):* The crystalline sample is set to Bragg-diffract the large, nearly parallel, monochromatic X-ray beam and a series of X-ray diffraction topographs is recorded along the global rocking curve using a pixelated two-dimensional detector. Each pixel records an individual rocking curve originating from a defined sample volume. Real-space maps are generated from the ensemble of the recorded rocking curves: by fitting to a Gaussian profile, these maps allow for a quantitative investigation of local crystalline distortions.

The experiments were carried out in transmission geometry at the beamline BM05 at the European Synchrotron Radiation Facility (ESRF), Grenoble, France. Beams of either 18 keV (λ = 68.88 pm) or 30 keV (λ = 41.33 pm) were extracted from the synchrotron radiation using a vertically diffracting Si (111) double-crystal monochromator. We combined the measurements with these different energies to study the influence of the absorption conditions on the defect images (Appendix A). All samples were mounted in such a way that the GaN $\bar{1}\bar{1}20$ or $\bar{1}100$ reflection was accessible in the vertical scattering plane. The beam size was 5 × 5 mm^2^. Bragg diffraction images were recorded on a detector system comprising a scintillator screen equipped with microscope folded-relay optics to a 2048 × 2048 pixel low noise Excelitas PCO *edge gold 4.2* sCMOS camera (Excelitas PCO GmbH, Kelheim, Germany). The use of this system resulted in an effective pixel size of 1.2 × 1.2 μm^2^, so the recorded images had a corresponding field of view of ~2.4 × 2.4 mm^2^. The distance between the sample and the detector was 50 mm. The sample was rotated along the rocking curve with an angular step of 10^−4^ degrees over a range of 0.01°. For every step, an image was recorded, i.e., a total of 100 X-ray diffraction images were captured for each scan. The generated stacks of images contain, for each of the pixels, a “local” rocking curve. These ~4 × 10^6^ rocking curves were processed by the Rocking Curve Imaging Analysis (RCIA) software tool [22]. Each of the rocking curves, corresponding to a single pixel, was fitted independently using a Gaussian function. Maps of the integrated intensity (INT), the full-width at half-maximum (FWHM), and the peak position angle (PPOS) of the diffracted intensity are then calculated from the ensemble of these rocking curves.

The INT map is comparable to a classical X-ray topograph and provides information on the defect structure of the crystals. The FWHM and the PPOS maps provide, in addition, quantitative values related to the local effective misorientation δΘ of the used lattice planes [9]. This effective misorientation is mainly constituted by the local variation in the rotation Δφ (or tilt) of these lattice planes and the local relative variation in the lattice parameter spacing Δd/d. These maps are, therefore, quantitative measurements of the local crystal lattice distortion and indicate the local lattice quality of the investigated crystal sample [23].

Under low-absorption conditions (µt < 1), both higher diffracted intensity and higher FWHM values correspond to an increased gradient of distortion within the diffracting volume. This is not what occurs for higher absorption conditions (typically µt > 6), when Borrmann conditions are present, as is the case for some of our experiments on Am-GaN, the INT map will behave reciprocally to the FWHM map, i.e., for areas of high defect density, the FWHM map will display high values, but the INT map will exhibit low intensity. This is because of the fact that defects “disturb” the anomalous transmission; when the strain gradient around the defect is high, new wavefields are created and propagate. Most of these new wavefields are strongly absorbed and this leads to a locally reduced intensity (“Borrmann shadow”) [9,24]. The PPOS map provides information on the local departure from the nominal Bragg angle. As indicated above, this departure includes contributions from lattice rotations (tilts) and from lattice plane spacing variations (strain) [23,25]. By combining the INT, FWHM, and PPOS maps, we can obtain extra quantitative information with respect to that provided by classical topography techniques.

The principle of the RCI technique and evaluation procedure are presented schematically in Figure 1. Table 1 gives an overview of the measurement conditions for all three XRT methods used for this study. The values were calculated using LauePT software [21].

### 2.3. Defect Selective Etching

Defect selective etching (DSE) was used as a complementary tool with respect to Bragg diffraction imaging. DSE is an established characterization technique for the analysis of defects in GaN thin films or bulk crystals. The method is suitable for revealing a wide range of different defects in GaN, such as dislocations, precipitates, grain boundaries, inversion domains, striations, and others. DSE is very sensitive to the local deformation of the GaN lattice and is mainly used to reveal dislocations. A particular advantage of the method is the relatively easy identification of the different observed types of TDs in the frequently used (0001) GaN crystals or substrates. Three types of TDs occur in (0001) GaN, namely, screw type (TSD) with Burgers vector ***b*** = 〈0001〉; edge type dislocation (TED), ***b*** = $1/3\langle 11\bar{2}0\rangle$; and mixed-type (TMD), ***b*** = $1/3\langle 11\bar{2}3\rangle$. When etching Ga-polar (0001) GaN, the etch pits differ significantly in terms of size, shape, and inclination angle of the side walls of the pits, depending on the doping and impurity level, the composition of the etchant, the etching temperature and time, and the type of the dislocations. However, for the three types of TDs in Ga-polar (0001) GaN, the size grading is generally such that TSDs form the largest etch pits, followed by TMDs, and TEDs form the smallest etch pits [26,27,28,29].

The approach of the DSE investigations in this work was to first use L-XRT to localize the HCDs on the Am-GaN wafers and then to apply target-oriented DSE at the relevant sites. For revealing dislocations, classical orthodox etching in molten E + M etch was used (eutectic mixture (E) of KOH + NaOH with 10% of MgO (M) [27]). In the present work, etching was performed on Ga-polar surface at a temperature of ~550 °C for 20 min. The temperature was measured inside a hot steel plate. After etching, the samples were examined by differential interference contrast (DIC) microscopy (Nikon Eclipse 80i). More details of the principle of DSE of GaN and other semiconductors can be found in the work of Weyher and Kelly [30].

## 3. Experimental Results

### 3.1. Results of X-ray Bragg Diffraction Imaging

#### 3.1.1. Tracking of Honeycomb Defects (HCDs) by Lang X-ray Topography (L-XRT)

Figure 2 shows L-XRT images of three 1-inch Am-GaN substrates recorded using the $11\bar{2}0$ reflection. The topographs in Figure 2a,b are from semi-insulating substrates and the topograph in Figure 2c is from an n-type one. We use the usual photographic convention, where higher X-ray intensity corresponds to increased blackening.

The experimental conditions described above coupled with the high crystalline quality of the crystals led to the fact that all the topographs were recorded under anomalous transmission (Borrmann effect) conditions. This means, as indicated above, that defects become visible through a locally reduced intensity. In the topographs of the Am-GaN substrates shown on Figure 2, different types of defects are visible on a µm-scale up to a mm-scale. The defects on the µm-scale are threading dislocations (TDs) of the screw (TSD), edge (TED), or mixed (TMD) type. In addition, macroscopic defects with mm-scale such as subsurface damage and scratches, traces of facets, growth bands, and various forms of dislocation clusters can be observed. The dislocation clusters can be divided into single locally isolated clusters; chains clusters; and a particular defect that consists of six clusters with a hexagonal, honeycomb-like, arrangement. These “honeycomb defects” (HCDs) constitute the topic of the present work. Descriptions and discussions regarding the other defect types visible in the overview topographs can be found elsewhere [8].

The HCDs we identified in the three substrates of Figure 2 are indicated by red hexagons. It can be seen that the HCD clusters display different sizes and distributions. In Figure 2b,c, the HCDs are rather isolated, whereas in Figure 2a, there is an area in the right part of the substrate where the HCDs are arranged close to one another. It should be noted that, from a statistical point of view, although the HCD is found in many Am-GaN substrates, the “density” of this defect is rather low; we observed, on the more than 40 studied substrates, a density between ~0.2 and 2.4 cm^−2^.

#### 3.1.2. Diffraction Imaging Contrast of the Honeycomb Defect (HCD) in Transmission L-XRT

Figure 3a shows the magnified cluster HCA-A of the overview topograph of Figure 2b (L-XRT, $11\bar{2}0$ reflection). The entire HCD cluster has a “diameter” of ~1.8 mm. Six areas of distinct light-dark contrast and hexagonal arrangement are observed, each forming large, many-lobed contrast rosettes. The arrangement of the six contrast rosettes is in a hexagonal shape and the outer edges of the honeycomb run along the *a*-plane $\left\{11\bar{2}0\right\}$ prismatic facets. Figure 3b shows the same topograph as Figure 3a with the six contrast rosettes numbered (1–6) as well as indexed planes for a better understanding of the following discussions in this paper. The contrast of the six rosettes does not look all the same, but there are three pairs of contrast rosettes with similar appearance, namely, (5, 6), (2, 3), and (1, 4), respectively. Using the orientation of the diffraction vector ***h*** and the numbering scheme of Figure 3b, the contrast rosettes can be described as follows: rosettes 5 and 6: two large lobes of dark contrast nearly perpendicular to ***h*** and two small lobes of bright contrast nearly parallel to ***h***; rosettes 2 and 3: two large lobes of bright contrast nearly perpendicular to ***h*** and two small lobes of dark contrast nearly parallel to ***h***. Rosettes 2 and 3 are a kind of contrast-inverted mirror image of rosettes 5 and 6 and rosettes 1 and 4 are an intermediate appearance of rosette types 5, 6 and 2, 3, but with much smaller dark and bright contrast lobes. Figure 3c shows magnified examples of each of the three different contrast rosette pairs. The size of the single rosettes exhibits a width of ~500 µm.

The six contrast rosettes of the HCD are embedded in a matrix containing isolated TDs. The TDD around the HCD shown here is ~5 × 10^4^ cm^−2^. Under Borrmann contrast imaging conditions, TDs become visible as many-lobed rosette-shaped contrasts in low defect crystals. Similar images have already been observed for TDs in highly absorbing semiconductors such as Ge [31], GaAs [32,33], CdTe [34], and ZnGeP_2_ [35,36,37]. Recently, many-lobed rosette-shaped contrasts of TDs were reported for low-defect Am-GaN for the first time [8]. The appearance of the many-lobed contrast rosettes of the HCDs is comparable to the contrast rosettes of single TDs in Am-GaN, but their dimensions are about 10 times larger. There is no doubt, however, that the defect images of large, many-lobed contrast rosettes observed here by L-XRT under high µt conditions correspond to a strong and very complex Borrmann effect of a specific defect cluster in GaN, suggesting a similarity in defect type of HCDs to TDs that was not tangible at this point.

For further discussion of the HCD contrast rosettes in transmission XRT, see Appendix A.

#### 3.1.3. Diffraction Imaging Contrast of the Honeycomb Defect (HCD) in Back-Reflection SWB-XRT

To obtain a simpler image of the HCD clusters, mainly by restricting to the surface, a SWB-XRT topograph was recorded in back-reflection geometry. Figure 4a shows the 0006 reflection topograph of the defect cluster labeled HCD-A in Figure 2b. Six contrasts displaying a hexagonal arrangement are clearly visible. The contrasts located at the six corners of the hexagon show two parts: a dark inner part and a bright, radially aligned, outer part. It has been shown that, using the SWB-XRT back-reflection technique, the threading dislocations (TDs) become visible as more or less point-shaped contrasts [38]. Three types of TDs, namely, screw, edge, or mixed, can be associated with varying size, brightness (light or dark), and uniformity of the contrast. The contrasts seen in Figure 4 are similar to what was observed before for TDs in Am-GaN crystals. The images we observe on the corners of the HCD could, therefore, result from the sum of the contrasts of aligned TDs.

Let us note that some of the six images display a single bright straight line (contrasts 1, 5, and 6 in Figure 4a), whereas others show a series of bright lines, suggesting a different structure of the associated defects (contrasts 2–4 in Figure 4a). The described features are easy to observe on the magnified image of the honeycomb corners 1 and 2 shown in Figure 4b. The corner 1 contrast exhibits a bright straight radial line, whereas in corner 2, the bright contrast splits into several nearly radial lines that, as will be shown in Section 3.2, correspond to several dislocation bundles.

In addition, we observe images of TDs present outside the corners of the HCD, which display a black dot–white dot contrast. The orientation of the line joining the center of the dots varies between TDs to the other, depending on their Burgers vector direction [39].

#### 3.1.4. Variations in the Honeycomb Defect Images

Although the HCDs observed by L-XRT in the different Am-GaN substrates show basically the same defect pattern, they display striking differences in the details of their image contrast. Figure 5 shows eight different HCDs, again marked with a red hexagon for better visibility. The occurrence of HCDs is independent of the doping type used in the substrates: the HCDs of Figure 5b–e were observed in n-type samples, whereas the HCDs of Figure 5a,f–h were found in semi-insulating (Mn-doped) substrates. The image size and contrast vary considerably from HCD to HCD. The sizes of the observed HCD vary from 50 µm to more than 2 mm. The rosettes present in the HCD images are not associated with the size; Figure 5a displays large and strong contrast rosettes, whereas Figure 5g shows small and weak contrasts, with both HCDs having a size of more than 2 mm. The contrast of the rosettes can be very different from one HCD to the other; Figure 5h shows, for instance, a HCD with a rather weak contrast. In addition to the six rosettes that form the hexagon, we observe, in some cases, other rosettes that are marked by small blue circles (Figure 5c–f), which occur both inside and outside the HCD. The hexagonal symmetry of the HCD cluster does not appear to be affected by these additional rosettes or by the TDs present in the crystal matrix. Figure 5e,f show, for instance, that, inside the HCD indicated by the red lines parallel to the *a*-planes, six additional rosettes form a second hexagon (marked in green), which is rotated by 30 degrees and bounded by the six $\left\{\bar{1}100\right\}$ *m*-planes. In addition, another rosette occurs exactly in the center of this defect cluster. Quite different types of HCDs can arise simultaneously on the same single wafer, so that no direct correlation to specific growth conditions can be found for the individual types of HCDs. As the distortion fields observed on the topographies vary between the HCDs, the features leading to the hexagonal defects may also vary within a general framework.

### 3.2. Results of Defect Selective Etching (DSE)

A total of seven HCDs of different sizes, from different Am-GaN substrates, and with different contrast details observed by L-XRT, as described in Section 3.1.4, were investigated by DSE. There are various types of HCDs; examples with the typical differences will be described in detail below.

Figure 6a shows the DIC optical overview image after orthodox etching of the HCD-A (Am-GaN substrate from Figure 2b). The use of DSE allows to identify the defects responsible for the hexagonal pattern of six multi-lobed contrast rosettes observed on Bragg diffraction images; these are bundles of TDs that are concentrated in the corners of the honeycomb in very restricted locations (marked by blue circles).

The analysis of the pit size and morphology of the pits indicates that these are TEDs. Figure 6b shows scaled-up images of the etch pits of the six dislocation bundles (DBs). The number of edge dislocations present per DBs is high, amounting to 127–199 individual TEDs (average number: 154 TEDs). There are also differences in the distribution of the TEDs in the DBs, with two different groups. The DBs labeled 1, 5, and 6 consist of TEDs that line up predominantly along single, straight long chains (~150 µm in length) and only a few single TEDs are located next to these chains. In a similar way, the edge dislocations of DBs 2, 3, and 4 are again predominantly arranged in several, nearly parallel, but shorter chains with a length of ~50 µm. The dislocation chains are radially oriented and parallel to one of the six crystallographic $\langle \bar{1}100\rangle$ directions. Within the chains, the distance between TEDs varies between 0.5 µm and 2 µm. DSE shows, inside and outside of the hexagon, other pits associated with further TDs, which were already mentioned when describing the L-XRT and SWB-XRT results. Besides TEDs, there is a large number of TMDs and few TSDs in the matrix. The TMDs and TSDs differ from the TEDs in the DSE analysis by forming larger etch pits. In some cases, TDs bundle together in the matrix. However, these are different from the six DBs that form the HCD cluster. The number of TDs is much lower, with around ~10–30 dislocations per bundle and, besides TEDs, TMDs also occur in these bundles. Some of these additional DBs are marked by yellow circles or ellipsoids in Figure 6a.

We pointed out in Section 3.1.4 that the individual HCDs display quite different details in terms of size, contrast strength, contrast of the dislocation images associated with the corners of the hexagon, and extra images associated with other dislocations. These differences in the XRT images led to extending the DSE analysis to other HCDs. Figure 7 shows a DIC optical image of the etched HCD shown in Figure 5c; etch pits of the six DBs of the TED type are again observed, but their number now ranges from 47 to 63 per DB, with an average number of 53. These are arranged, within the bundles, in several short chains with lengths of 10–20 µm. The L-XRT image (Figure 5c) showed another rosette in the center of the hexagon. DSE reveals that these are also pits from a large bundle of TDs (labelled as 7 in Figure 7). The pits of this central DB are larger than the ones forming the HCD, this indicating that they are TMDs. Further smaller groups of clustered TMDs are also present in the interior of the HCD hexagon. We can also observe in the interior of this HCD two large deep pits formed on screw dislocations that appear in the center of former growth hillocks (TSD-1 and TSD-2, Figure 7a,b). To complete the description of the DSE analysis of this sample, we also observe defects related to growth facets (F) and scratches (S).

Figure 8 shows, as a third example, the DIC optical image after etching of the HCD shown in Figure 5h, which was detected in a semi-insulating Am-GaN substrate. Here, the etch pits corresponding to TEDs in the six corners of the honeycomb are spread over a large area (with a “diameter” of ~250 µm, Figure 8b) and show almost no chain-like arrangement. The number of TEDs per DB is between 80 and 120 (average: 101). The count of TEDs for this HCD is not so precise owing to the wide dispersion of the distribution.

For the seven HCD clusters analyzed by DSE, it can be stated that, in each case, the six DBs forming the hexagon are always composed of pure TEDs. On the other hand, DBs inside and outside the HCD cluster are mostly bundles of TMDs, but some TED bundles were also observed. Pure TSDs were not observed as bundles; these occurred only in isolation.

The examples of HCDs examined here using DSE show the following characteristics:(i)single TEDs in the DBs arrange as a few (or even single) long chains, many short chains, or are distributed over an area of up to 250 µm. There appears to be a correlation between the way the TEDs are located and the images they produce on the topographs: when more TEDs in the DBs are arranged as chains, stronger contrast is recorded on L-XRT;(ii)the number of TEDs in the DBs differs significantly between HCDs. We can extract a correlation, shown in Figure 9, between the number of TEDs in the DBs and the size of the HCD clusters: the higher the number of TEDs in the corners of the hexagon, the larger the size of the corresponding HCD cluster.

### 3.3. Use of RCI for a Quantitative Determination of the HCD´s Associated Lattice Distortion

The synchrotron RCI measurements were performed on the HCD-A indicated in Figure 2b, which was also investigated previously by L-XRT, SWB-XRT, and DSE. For these studies, the $\bar{1}\bar{1}20$ reflection and $\bar{1}100$ reflection of the prismatic GaN planes were used. In order to try to facilitate the image interpretation, we decided to mainly use a 30 keV energy incident X-ray beam (µt ~1.9, 30 keV), which allows to avoid being in Borrmann conditions. A comparison of the images recorded under high absorption (µt~7.9, 18 keV) and the medium absorption displayed here is shown in Appendix A.

From the images recorded along the rocking curve, as already indicated, maps of intensity (INT) and, with the help of Gaussian fits of the rocking curves recorded in each individual pixel, maps of full-width at half-maximum (FWHM) as well as peak position (PPOS) were extracted.

The first, simple approach to gain information on the DBs forming the HCD cluster can be achieved through the evaluation of the original individual topographs recorded along the rocking curve for the 30 keV measurement of the $\bar{1}\bar{1}20$ reflection. This enables to observe “simple” images of defects because, if an image is recorded at an angular position ΔΘ very far away from Θ_centre_, the central position of the diffraction curve, the main contribution to Bragg diffraction originates from the distorted regions lying around the defect, and not from the perfect-crystal matrix. This XRT variant is termed “weak beam topography” [40,41].

Figure 10 shows a series of five individual images taken along the rocking curve for the area of HCD-A (see Figure 2b). The image shown in Figure 10c was taken in the center of the rocking curve (Θ_centre_); images from Figure 10b,d in the steeply rising and steeply falling flanks of the rocking curve, respectively; and images from Figure 10a (left tail) and Figure 10e (right tail) far away (ΔΘ~ ±60 µradian) from the maximum Θ_centre_ of the rocking curve. The left tail image (Figure 10a) and right tail image (Figure 10e), respectively, show the weak beam images of the six DBs of the HCD-A. These topographs were taken outside the exact Bragg conditions and, therefore, no dynamical-theory related contrast is visible. Instead, only a “direct contrast image” of the lattice distortion associated with the DBs is observed. Further analysis of the original individual INT images captured along the rocking curve shows that the areas inside and outside the HCD become visible on different topographs (in particular Figure 10b,d). This indicates that the Am-GaN crystal regions inside and outside the honeycomb are in diffraction position for different angles, probably corresponding to a small misorientation.

Figure 11 and Figure 12 show the RCI maps at 30 keV of the $\bar{1}\bar{1}20$ and $\bar{1}100$ reflections, respectively. The INT maps presented in Figure 11a and Figure 12a are, in spite of the difference in absorption, qualitatively similar to the topographs of the L-XRT and SWB-XRT experiments. Indeed, the HCD shows in the INT maps six large multi-lobed rosettes, which appear to be the images of the DBs composed of 127–199 individual TEDs. The formation of these distinctive contrast rosettes is, as already indicated, associated with dynamical theory-related interferences that produce high-structural-quality crystals. These interferences also show up, in a reverse way, in the FWHM map, but it can be observed that, in addition, the DBs located in the six corners of the hexagon exhibit a higher FWHM (10–15 µradians) than the one of most of the pixels in the image (~5 µradians). The PPOS map shows, around most of the corners of the hexagon, a blue-red contrast corresponding to a variation in peak position of ~30–40 µradians. It is known that a variation in peak position appears around a single dislocation and its value (typically 5 µradians) is proportional to the Burgers vector. Tsoutsouva et al. [25] have reported that, for the case of parallel dislocations located within one small area (a few µm^2^ range), misorientation, measured a few µm away from this area, corresponds to an effective Burgers vector (the sum of the Burgers vector of the individual dislocations). In our case, this would lead to the conclusion that the sum of the Burgers vectors of the more than one hundred dislocations present on each of the corners is less than ten times the one of an individual dislocation. However, this procedure does not correspond to our case, because the dislocations present in the corners of HCDs are either arranged in “walls” or are located within relatively large areas (~ 200 µm).

We retrieve, in these PPOS maps, the variation in orientation, already observed in the images taken along the rocking curve (Figure 10), outside and inside of the HCD cluster. The mutually misoriented crystal volumes are separated by the formation of a kind of subgrain boundary lying along the notional connecting lines between the six DBs of the HCD cluster. Therefore, these subgrain boundaries run along the *a*-plane $\left\{11\bar{2}0\right\}$ prismatic planes. A clear complete hexagon cannot be observed for each diffraction vector used, but, depending on the orientation with respect to the diffraction vector, only certain subgrain boundaries of the honeycomb become visible.

Figure 13a,b show the PPOS maps of the $\bar{1}\bar{1}20$ and the $\bar{1}100$ reflection, respectively, with superimposed indications of the extra features we wish to point out. For better understanding, a scheme of the observed features and the corresponding crystallographic orientations is shown in Figure 13c. For the PPOS map corresponding to the $\bar{1}\bar{1}20$ reflection (Figure 13a), the subgrain boundaries of the $\left(2\bar{1}\bar{1}0\right)$, $\left(\bar{2}110\right)$, $\left(1\bar{2}10\right)$, and $\left(\bar{1}2\bar{1}0\right)$ facets forming the hexagon are clearly visible, whereas for the PPOS map of the $\bar{1}100$ reflection (Figure 13b), the subgrain boundaries of the $\left(11\bar{2}0\right)$ and $\left(\bar{1}\bar{1}20\right)$ facets stand out. The visibility of the subgrain boundaries is significantly reduced when they are perpendicular to the direction of the diffraction vector. This indicates that the rotation of these planes one side and the other of the subgrain boundary is such that the rotation axis has a main component parallel to the diffraction vector. From the observations, it follows that the misorientation of the volume is not preferentially in one of the six *a*-plane directions, but it is a three-dimensional misorientation. Let us note, however, that the average misorientation of the hexagon area relative to the surrounding Am-GaN matrix is small; the measurement of the average position inside and outside the honeycomb shows a variation of ~3 µradians for the $\bar{1}\bar{1}20$ reflection and ~8 µradians for the the $\bar{1}100$ reflection. We can also observe that, except for the large six DBs in the corners of the hexagon with the high numbers of TDs, almost no TDs occur on the boundary courses of the subgrain boundaries between the DBs.

In addition to the subgrain boundaries along the *a*-plane prismatic planes forming the hexagon (black dashed lines), the HCD cluster appears to be subdivided into sub-subgrains with boundaries (indicated by orange and violet dashed lines in Figure 13a–c), which mostly go from one TD to the other. These TDs act as geometrically necessary dislocations (GNDs; see, for instance, [42,43]) and can even form, as shown in Figure 13a–c, smaller hexagons. Most of the inner sub-subgrain boundaries run along the *a*-plane prismatic planes, but a few of them run along the *m*-plane $\left(\bar{1}100\right)\;$ prismatic plane (marked with a dashed violet line). These “sub-subgrains” within the HCD cluster exhibit very weak misorientations, in the µradian range, smaller than that of the entire HCD cluster relative to the surrounding Am-GaN matrix.

Figure 14 shows the magnified $\bar{1}\bar{1}20$ and $\bar{1}100$ PPOS map images of two opposite DBs of the HCD cluster, together with the corresponding DSE images, which indicate the occurrence of what we call a “limited size subgrain boundary”. Indeed, the edge dislocations that are aligned (Figure 14f) can, if they have the same Burgers vector (or at least when one Burgers vector is predominant), act as GNDs to produce a rotation of the lattice planes on one side of the alignment with respect to the other side. This variation in orientation is visible in Figure 14e as a red-blue contrast. However, in Figure 14d, we can notice that the contrast is actually more complicated, with a succession, within a globally “light blue” matrix, of “light blue–blue–red–blue–light blue” angular positions. This is expected for a limited size subgrain boundary because the induced misorientation must come back to the matrix orientation when far from the subgrain, implying a rotation in the opposite sense, as schematically indicated in Figure 15. This configuration is confirmed by the weak beam images of the RCI (Figure 10a,e), which show a double point on each of the corners, indicating that there are two areas (the red or blue ones in Figure 15 schematic drawing) around the limited size subgrain boundary displaying the same orientation.

We can, in addition, infer from Figure 14 that the aligned or partially aligned TEDs (Figure 14f,c, respectively) that constitute the DBs located in the corners of the hexagon induce qualitatively similar distortions. However, the limited subgrain boundary behavior schematically presented in Figure 15 is clearly visible only for the aligned dislocations. On the other hand, the Burgers vectors of the dislocations located in the opposite corners of the hexagon do not appear to be aligned. If they were aligned, the contrast of the DBs images would be the same (or reversed if the Burgers vector is reversed) in Figure 14a,d, which is not observed.

## 4. Discussion

Bragg diffraction imaging and DSE analyses revealed that all of the HCDs occurring in Am-GaN have a series of common features. Indeed, they are mainly constituted by six dislocation bundles, each consisting of edge-type threading dislocations (TEDs) with dislocation lines nearly parallel to the 〈0001〉 direction. The number of TEDs present in each corner varies between HCDs, but the higher this number, the bigger the HCD size. This size is a function of the number of edge dislocations, but their arrangement (lines, succession of small lines, more diffuse cluster) or the number of other type of dislocations in their neighbourhood appear to have a reduced influence.

The origin of the formation of the HCDs during ammonothermal GaN growth is not clear at the present stage. However, information can be provided on the stage of formation within the crystal growth process chain for these defect clusters. A complex technology is used to produce large area Am-GaN crystals, suitable for substrates with (0001) orientation, from small seeds. Process steps that have to be performed include the following: (1) multiple regrowth of slender GaN seed in the lateral $\langle 11\bar{2}0\rangle$ directions; (2) joining two crystals by tiling technology; (3) continuing GaN grow in the lateral $\langle 11\bar{2}0\rangle$ directions as well as in the vertical $\left[000\bar{1}\right]$ direction in multiple regrowth steps to a size sufficient for substrate fabrication; and (4) multiple regrowth in the vertical $\left[000\bar{1}\right]$ direction (with lateral growth blocked to increase only the thickness in the vertical $\left[000\bar{1}\right]$ direction) [5,6]. As has recently been shown, the history of the grown crystals, or more precisely, the individual process steps of crystal growth used for preparing these Am-GaN samples, can be traced in detail on the basis of observed XRT defect patterns [8]. Macroscopic defects, such as growth bands and chains of threading dislocations, related to regrowth steps as well as chains of threading dislocation bundles related to tiling seams of the joining process of two seed crystals for crystal area enlargement act as chronological markers of the Am-GaN growth. Figure 16 shows an HCD formed on chains of threading dislocation bundles (marked by red arrows) of a tiling seam. This observation is clear evidence that, chronologically, this HCD was formed after joining two seed crystals, i.e., in a process step that served to increase the thickness in the vertical $\left[000\bar{1}\right]$ direction. It can thus be ruled out that the HCD cluster formed in a process step for lateral crystal enlargement during growth in the $\langle 11\bar{2}0\rangle$ directions. Further research is needed in order to explain the reason for the formation of the HCDs.

In the case of extensively studied HCD-A, the subgrain boundaries form along the $\left\{11\bar{2}0\right\}$ prismatic planes and the total crystal volume within the hexagon is slightly misoriented relative to the surrounding Am-GaN matrix. The average misorientation of the investigated HCD-A volume is very small. At this stage, it is not possible to say if this low misorientation is caused by the DBs of the HCD. The origin can also be connected to isolated threading dislocations or other DBs, which do not belong to the six DBs forming the honeycombs. In addition, the volume of the hexagon is intersected by further sub-subgrain boundaries with even smaller misorientations.

We do not have, presently, a convincing model to explain the HCDs and further work is in progress to determine the Burgers vectors of the observed TEDs. Notwithstanding, from the observed misorientation of the bounded subgrain boundaries in the RCI PPOS maps, we can conclude that the Burgers vectors ***b*** in opposite corners cannot be collinear. On the other hand, no contrast is observed for subgrain boundaries lying along a side of the honeycomb when the diffraction vector ***h*** is perpendicular to the boundary. If we assume that the subgrain rotation originates from the DBs that are present in the corners of the HCD, which act as GNDs, this non-visibility could correspond to an “effective Burgers vector” in the middle of this subgrain boundary perpendicular to the diffraction vector ***h***. This effective ***b*** can be approximated by the sum of two effective ***b*** corresponding to two DBs in the neighboring corners, providing further clues for possible Burgers vectors directions. However, all of these considerations remain highly speculative and more information is required for preparing a model. We can nevertheless say that HCDs should result from attractive and repulsive interactions between the TED bundles sitting in various hexagon corners. Additionally, these interactions should have a different dependence on the size of the hexagon in order to have a minimum energy size, which is related to the number of dislocations in each corner.

## 5. Conclusions

In this paper, we analyzed defect clusters with hexagonal honeycomb-like arrangement in ammonothermally grown GaN substrates. These defect clusters vary in size from 0.05 mm to 2 mm. In each of the hexagons´ corners, we find a large concentration of edge-type threading dislocations forming bundles. Within these bundles, the dislocations are either grouped in areas or they show up as straight long chain-like radial alignments of the same size that behave like limited subgrain boundaries. No defect concentration is seen at the edges of the hexagons. The size of the honeycomb defect clusters correlates with the number of dislocations in the bundles. At present, no clear model for the formation of this complex defect type in GaN can be given. Further work is in progress to collect more statistics for these defect clusters as well as to determine by transmission electron microscopy the direction of the Burgers vectors of the edge-type threading dislocations in the bundles forming the geometrically necessary dislocations. Understanding and, possibly in future, preventing the development of HCDs during ammonothermal GaN crystal growth is of great importance, both because Am-GaN crystals are presently used to manufacture electronic or optoelectronic devices and because they are used as seed crystals for new generations of GaN bulk crystals, e.g., by ammonothermal or hydride vapor-phase epitaxy (HVPE) growth processes.

## Figures and Tables

**Figure 1 materials-15-06996-f001:**
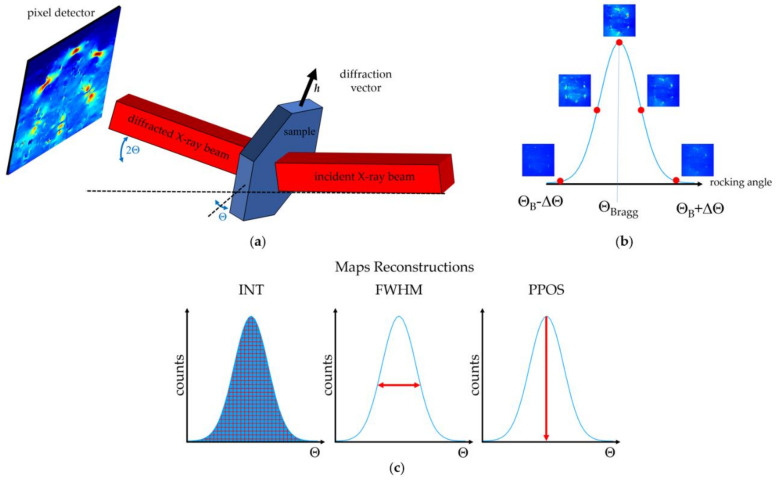
Schematic drawing of Bragg diffraction imaging using (**a**) rocking curve technique with a wide monochromatic incident beam; (**b**) acquisition of an image series (each pixel records its own local rocking curve); and (**c**) reconstructions of maps of the integrated intensity (INT), the full-width at half -maximum (FWHM), and the diffraction peak position (PPOS), extracted on a pixel-by-pixel basis from the local rocking curves.

**Figure 2 materials-15-06996-f002:**
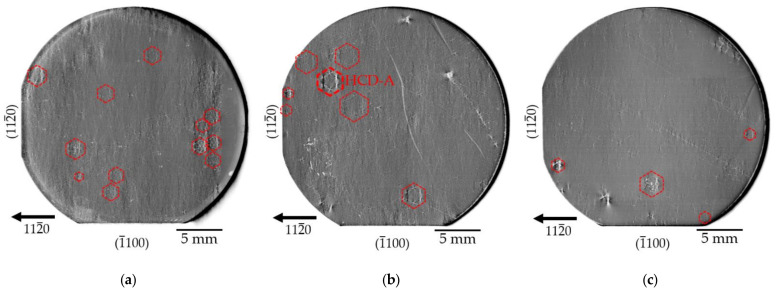
L-XRT $11\bar{2}0$ reflection topopgraphs of three GaN substrates with HCDs (marked by red hexagons); (**a**) and (**b**) are topographs of semi-insulating Am-GaN substrates, while (**c**) is a topograph of an n-type Am-GaN substrate. The defect cluster denoted as HCD-A in Figure 2b is subject of detailed studies in the following and is, therefore, highlighted.

**Figure 3 materials-15-06996-f003:**
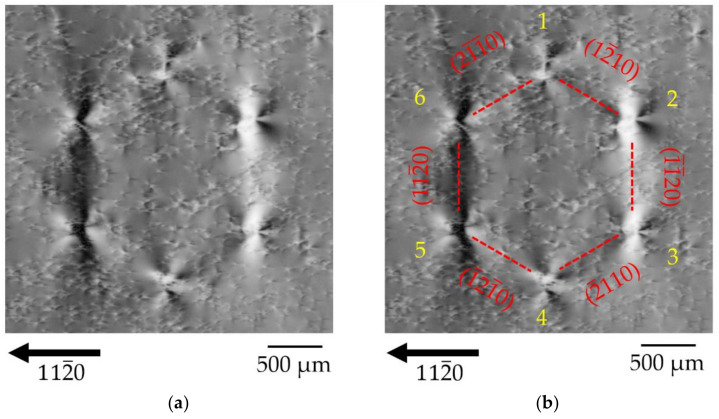
The HCD cluster shown in (**a**) is the enlarged L-XRT image, $11\bar{2}0$ reflection, of the one labeled HCD-A on Figure 2b; (**b**) for a better understanding of the explanations in the text, the contrast rosettes were numbered (1 to 6) and the crystallographic planes are indicated; (**c**) details of some of the corners of the hexagon are shown.

**Figure 4 materials-15-06996-f004:**
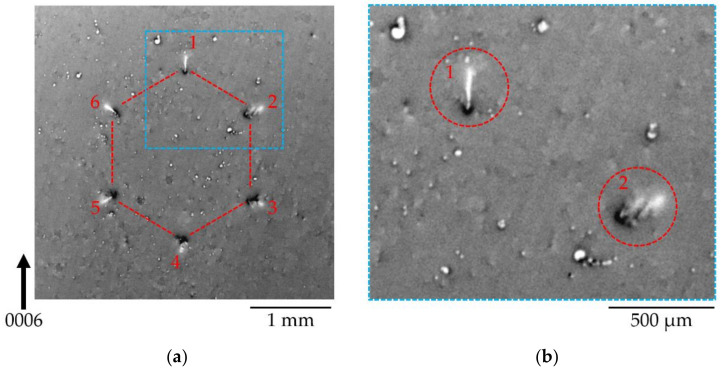
Back-reflection geometry 0006 SWB-XRT topograph showing (**a**) the hexagon labeled HCD-A in Figure 2b; (**b**) an enlarged view of corners 1 and 2.

**Figure 5 materials-15-06996-f005:**
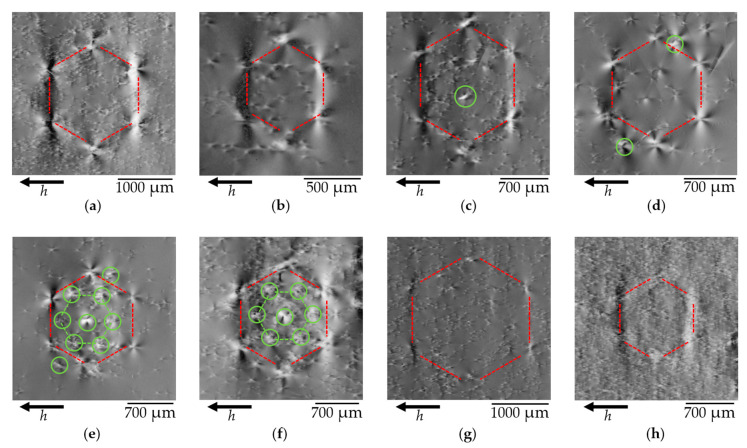
L-XRT of eight different HCDs (marked with a red hexagon for better visibility), which exhibit different sizes and different contrasts with light and dark areas arranged hexagonally, as well as often large many-lobed rosettes. All topographs were measured using a type $11\bar{2}0$ reflection. Additional clusters of threading dislocations are indicated by blue circles; the green hexagons mark a second kind of honeycomb bounded by the six $\left\{\bar{1}100\right\}$ *m*-planes. (**a**) HCD with strong contrast rosettes in semi-insulating Am-GaN, (**b**) HCD with strong contrast rosettes in n-type Am-GaN, (**c**–**f**) HCDs with in addition to the six rosettes that form the hexagon, other rosettes marked by small blue circles, (**e**,**f**) HCDs with six additional rosettes that form a second hexagon (marked in green), (**g**,**h**) HCDs with weak contrast rosettes in semi-insulating Am-GaN.

**Figure 6 materials-15-06996-f006:**
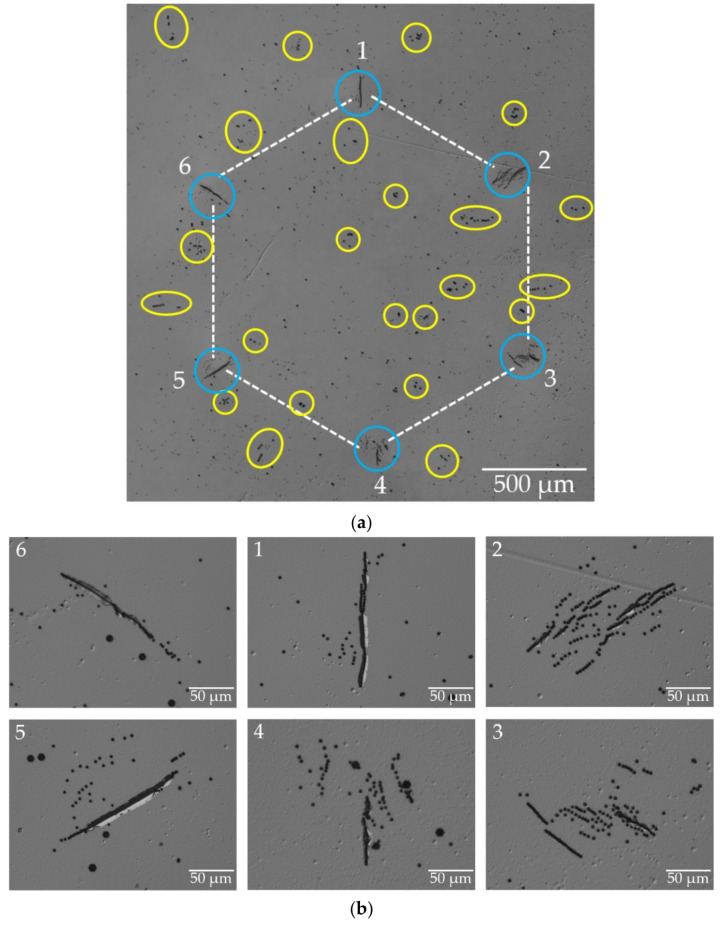
DIC optical images recorded after etching the HCD-A present in the Am-GaN substrate shown in Figure 2b: (**a**) overview image that allows identifying the bundles of TEDs that are concentrated in the corners of the honeycomb in very restricted locations (marked by blue circles); (**b**) enlarged images of the etch pits of the six dislocation bundles (DBs) located in the corners of the hexagon (a highly magnified image of the etched DB 5 is shown in Appendix B, Figure A4).

**Figure 7 materials-15-06996-f007:**
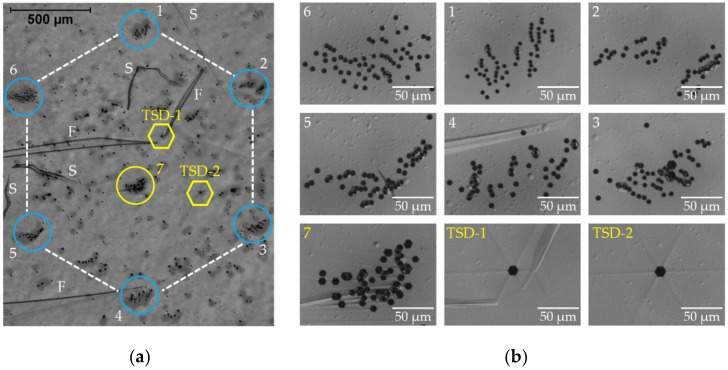
(**a**) DIC optical image of the etched HCD shown in Figure 5c. The corners of the hexagon are formed by TEDs. Growth facets (F) and scratches (S) are also visible. (**b**) Enlarged images of the etch pits formed on the bundles of TEDs (1-6). Other pits of dislocation bundles (like the one labeled 7) are of mixed type and two pits of single threading screw dislocations (TSD-1 and TSD-2) are observed.

**Figure 8 materials-15-06996-f008:**
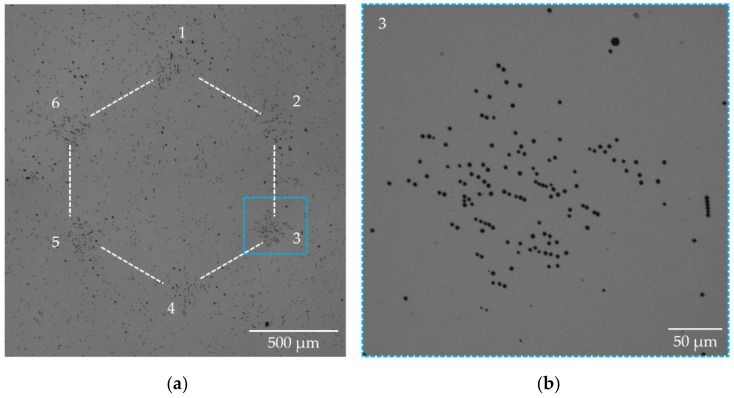
(**a**) DIC optical image of the etched HCD shown in Figure 5h. TEDs of each DB in six corners of the honeycomb cluster are spread over an area with a “diameter” of ~250 µm, as shown for DB-3 in the enlarged image (**b**).

**Figure 9 materials-15-06996-f009:**
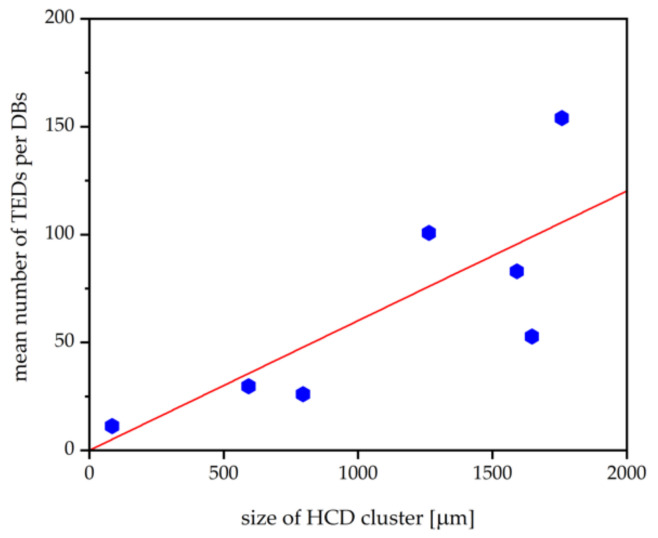
Correlation between the number of TEDs in the DBs located in the corners of the HCD and the size of the HCD clusters: the larger an HCD cluster, the higher the number of TEDs in the corresponding six DBs. The blue hexagons represent the data points and the red line is a guide for the eyes.

**Figure 10 materials-15-06996-f010:**
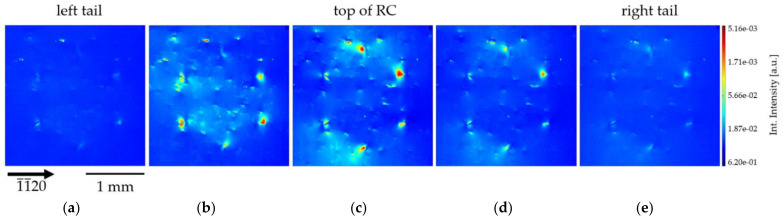
A series of individual images recorded on the HCD-A (from Figure 2b) for several angular positions along the rocking curve: going from its left tail, through the top, and to its right tail. The detailed description of the individual images (**a**–**e**) is given in the text. (**c**) image taken in the center of the rocking curve (Θ_centre_), (**b**,**d**) images from the steeply rising and steeply falling flanks of the rocking curve, (**a**) image from the left tail and (**e**) image from the right tail far away (ΔΘ~ ±60 µradian) from the maximum Θ_centre_ of the rocking curve, respectively.

**Figure 11 materials-15-06996-f011:**
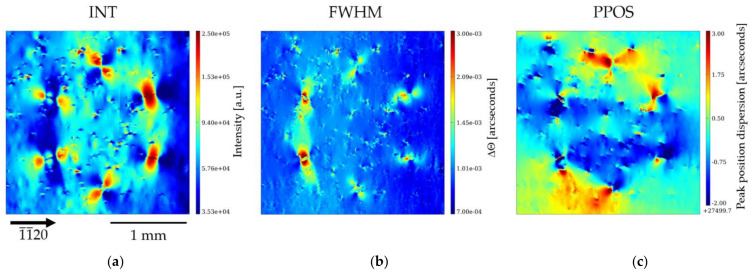
RCI maps of the $\bar{1}\bar{1}20$ reflection at 30 keV: (**a**) integrated intensity map (INT); (**b**) full-width at half-maximum map (FWHM); (**c**) peak position map (PPOS).

**Figure 12 materials-15-06996-f012:**
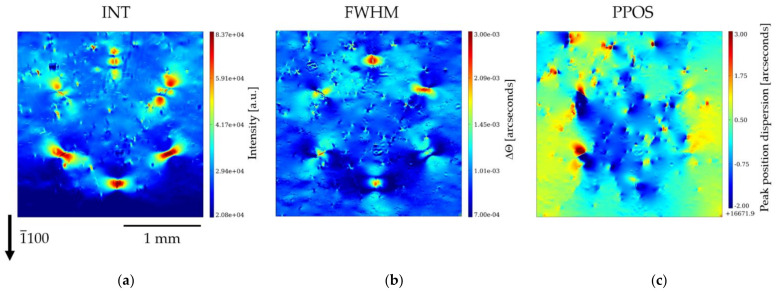
RCI maps of the $\bar{1}100$ reflection at 30 keV: (**a**) integrated intensity map (INT); (**b**) full-width at half-maximum map (FWHM); (**c**) peak position map (PPOS).

**Figure 13 materials-15-06996-f013:**
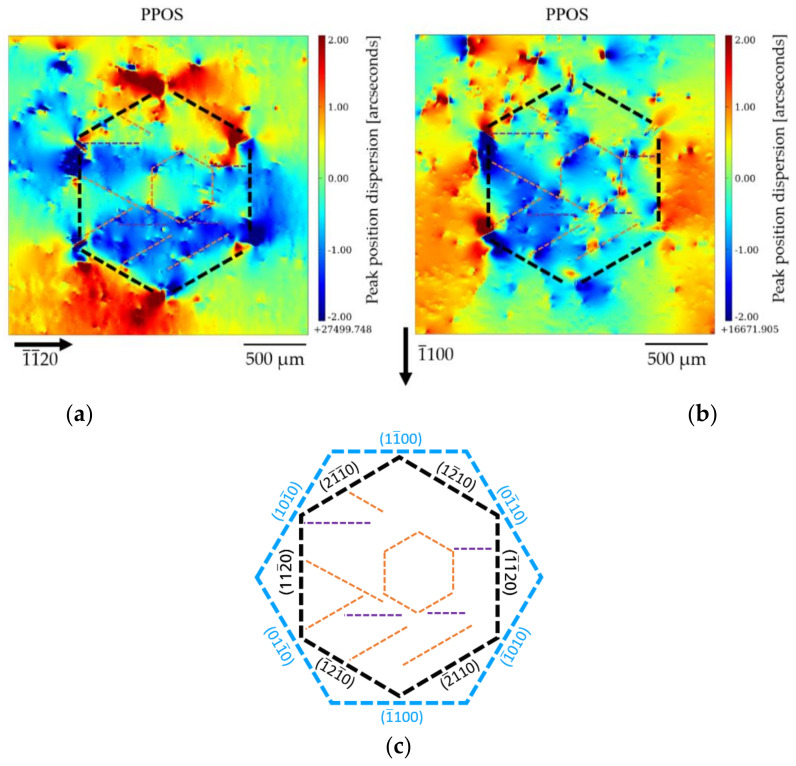
Peak position maps for (**a**) the $\bar{1}\bar{1}20$ reflection and (**b**) the $\bar{1}100$ reflection at 30 keV. The subgrain boundaries along the *a*-plane $\left\{11\bar{2}0\right\}$ prismatic planes forming the hexagon are indicated as black dashed lines and the subgrains present within the HCD are indicated as orange and violet dashed lines. These last subgrain boundaries mostly go from one TD bundle to the other and can even form smaller honeycombs within the big one. The misorientations among these subgrains are extremely small, in the very few µradian range. (**c**) Stand-alone subgrain boundaries shown. For better understanding, the orientations of the *a*-plane facets (black) and *m*-plane facets (blue) are shown.

**Figure 14 materials-15-06996-f014:**
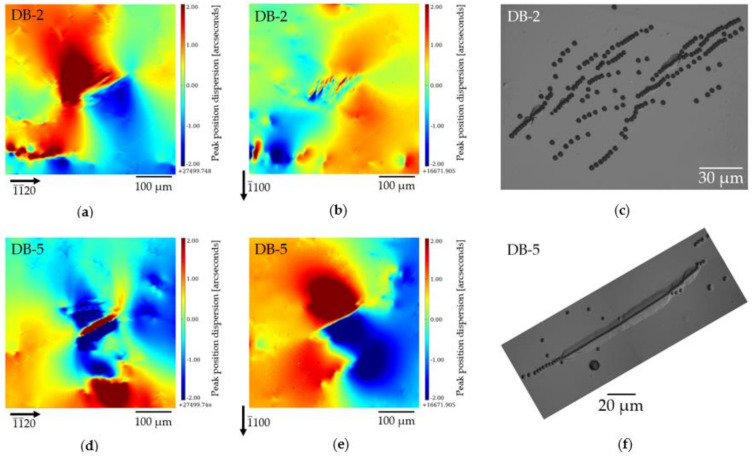
Magnified $\bar{1}\bar{1}20$ and $\bar{1}100$ PPOS map images of two DBs (indicated as 2 and 5 in, e.g., Figure 3b and Figure 6a) of the HCD-A cluster, together with the corresponding DSE images (a highly magnified image of the etched DB 5 (Figure 14f) is shown in Appendix B, Figure A4). The explanation for the individual figures is given in the text. (**a**) $\bar{1}\bar{1}20\;$ reflection PPOS map of DB-2, (**b**) $\bar{1}100$ reflection PPOS map of DB-2, (**c**) DSE image of DB-2, (**d**) $\bar{1}\bar{1}20$ reflection PPOS map of DB-5, (**e**) $\bar{1}100$ reflection PPOS map of DB-5, (**f**) DSE image of DB-5.

**Figure 15 materials-15-06996-f015:**
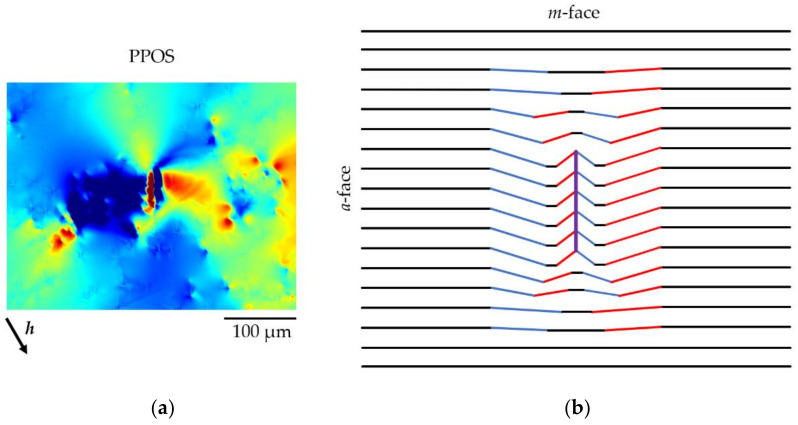
Lattice plane rotations around a limited size subgrain boundary; (**a**) RCI PPOS measurement of the lattice rotation; (**b**) schematic drawing: black lines—no rotation of the lattice planes, blue lines—downward rotation of the lattice planes, red lines—upward rotation of the lattice planes, and purple line—limited size subgrain boundary.

**Figure 16 materials-15-06996-f016:**
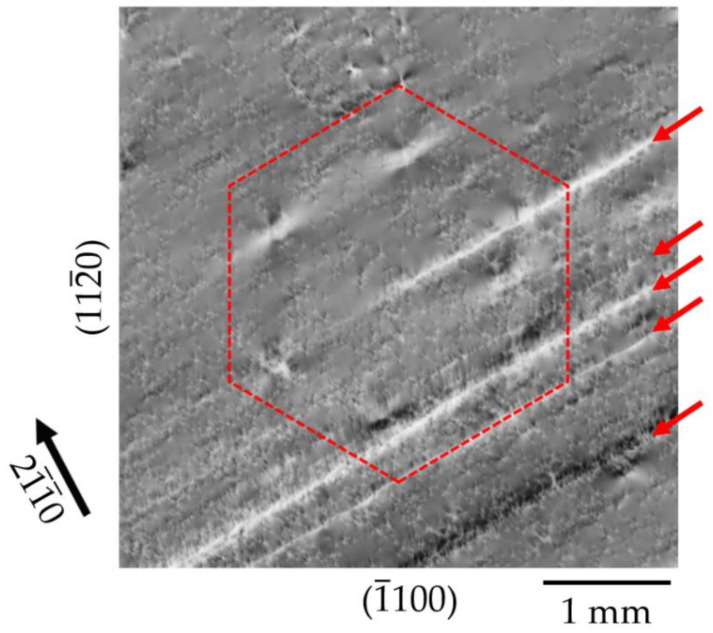
HCD (marked with a hexagon) formed on chains of threading dislocation bundles (red arrows) of a tiling seam. This allows a chronological assignment of the development of this defect cluster in the stages of seed enlargement.

**Table 1 materials-15-06996-t001:** Overview of the used measurement conditions for L-XRT, SWB-XRT, and RCI.

Technique	Reflection hkil	Energy [keV]	Wavelength λ [pm]	Bragg Angle Θ_Β_ [°]	Absorption µt *
L-XRT	$11\bar{2}0$	8.05 (Cu-Kα_1_)	154.06	28.89	9.1–13.6
SWB-XRT	$11\bar{2}0$	18.70	66.30	12.00	7.1
	$\bar{1}100$	12.47	99.45	10.37	20.5
	$11\bar{2}0$	18.00	68.88	12.47	7.9
RCI	$11\bar{2}0$	30.00	41.33	7.45	1.9
	$\bar{1}100$	30.00	41.33	4.29	1.9

* µ = linear X-ray absorption coefficient, t = thickness of GaN substrate, - thickness of the GaN substrates investigated by L-XRT using Cu-Kα_1_: 300 µm–450 µm, - thickness of the GaN substrate with the intensively investigated defect designated “HCD-A” that was the subject of detailed investigations using L-XRT (Cu-Kα_1_) and, in addition, SWB-XRT as well as RCI: 300 µm.

## Data Availability

The data presented in this study are available upon request from the corresponding author when reasonably indicated.

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
