# Peer review of "Large-Scale Defect Clusters with Hexagonal Honeycomb-like Arrangement in Ammonothermal GaN Crystals"

_materials, 2022, doi:10.3390/ma15196996_

Round 1

Reviewer 1 Report

Manuscript ID: materials-1919230

Referee Comments:                                                                                      Date: 09/15/2022

This manuscript by Lutz Kirste et al., entitled as “Large‐scale Defect Clusters with Hexagonal Honeycomb Like Arrangement in Ammonothermal GaN Crystals” presents the study of honeycomb like defect cluster in Am-GaN crystals using Bragg diffraction imaging techniques in combination with defect selective etching. Authors also observed the defect cluster is dependent on Am-GaN crystal size. The subject is interesting and could be useful for real time applications. However, there are some issues in this study and its presentation. The manuscript can be considered for publication in “materials” after minor revision. The detailed comments are listed as follows:

Abstract and Introduction:

1.     In line 322-325, “many‐lobed rosette‐shaped contrasts of TDs were reported for low defect Am‐GaN…………………………………….. single TDs in Am‐GaN, but their dimensions are about 10 times larger.” How is this study is different over previous study (in Ref. #8) in terms of type of defects, other than the size?

2.     Correct the gramaatical mistake in line 594.

3.     What are the factors affecting the defects in Am‐GaN crystals during the synthesis process?

Reviewer 2 Report

All aspects related to obtaining perfect GaN crystals are extremely important for modern science and technology. Therefore, the topic of the work is very relevant. The authors are the first to describe accumulations of defects with a hexagonal like arrangement (HCDs) in ammonothermally grown GaN substrates.

Perhaps, indeed, the phenomenon of the formation of a macroscopic ordered defect structure in GaN - “honeycomb defect” (HCD) - has not been described anywhere before and is observed by the authors for the first time.

In the submitted manuscript, this phenomenon is described quite exhaustively with the involvement of modern techniques and submitted for discussion.

The experiment was performed at a high level and described clearly and clearly. The illustrations are well designed and informative. Abstract and conclusions reflect the content of the work.

Unfortunately, the authors cannot explain the reason for the formation of the HDSs (I also do not find an explanation). Nevertheless, I consider it right and necessary to present this observation to the public.

It is difficult to judge the scientific significance of the result. Since the phenomenon concerns the strategically important GaN material, the information can be classified as significant on this basis alone. Moreover, the observed phenomenon with a high degree of probability is of a general nature for crystals of various substances. But the causes of HDSs remain unclear, and the authors themselves do not write anything about their possible significance.

As for the list of references, one could refer to a considerable number of other publications, since many works have been devoted to the growth of crystalline GaN to date. However, I think it's copyright to choose the cited publications. They adequately cover the research topic. Therefore, I believe that the list of references does not need to be corrected.

The manuscript does not cause me any complaints, except for one typo on p. 594 - TThe.

English level of the manuscript I do not undertake to evaluate. Personally, I had no difficulty in understanding.

Reviewer 3 Report

The present work investigated the new type of extended defect that occurs in ammonothermally grown GaN single crystals using X‐ray Bragg diffraction imaging and defect selective etching. The authors reported that the observed size of the honeycomb ranges from 0.05 mm to 2 mm and is correlated with the number of dislocations located in each of the hexagons' corners: typically ~ 5 to 200, respectively. I found this manuscript suitable for publication after some minor revisions:

1- The necessity of doing this work should be presented in the first sentence of the abstract section.

2- the future outline of this project should be stated in the last sentence of the abstract paragraph.

3- There are many keywords. I suggest using five keywords.

4- Using GaN abbreviation in the abstract section could be obscure for the journal readers in the abstract section. So, Please mention that GaN is the abbreviation of which material. 

5- Don't use bulk references.

6- The introduction section is too short. Use more literature for cooking your own story.

7- The results and discussions section is properly good written. So, there is no need to change this section.

8- in the conclusion section, the authors need to discuss future outlines of their findings in detail.
